# Type I Cystatin Derived from *Cysticercus pisiformis*—Stefins, Suppresses LPS-Mediated Inflammatory Response in RAW264.7 Cells

**DOI:** 10.3390/microorganisms12050850

**Published:** 2024-04-24

**Authors:** Qianqian Yang, Jia Li, Lilan Zhang, Ningning Zhao, Xiaolin Sun, Zexiang Wang

**Affiliations:** College of Veterinary Medicine, Gansu Agricultural University, Lanzhou 730070, China; yangqq@gsau.edu.cn (Q.Y.); lijia@st.gsau.edu.cn (J.L.); zhangll@st.gsau.edu.cn (L.Z.); zhaonn@st.gsau.edu.cn (N.Z.)

**Keywords:** *Cysticercus pisiformis*, stefin, anti-inflammatory effect, macrophage RAW264.7 cells, dose-dependent

## Abstract

Cysticercus pisiformis is a kind of tapeworm larvae of *Taenia pisiformis*, which parasitizes the liver envelope, omentum, mesentery, and rectum of rodents such as rabbits. Cysteine protease inhibitors derived from helminth were immunoregulatory molecules of intermediate hosts and had an immunomodulatory function that regulates the production of inflammatory factors. Thus, in the present research, the recombinant Stefin of *C. pisiformis* was confirmed to have the potential to fight inflammation in LPS-Mediated RAW264.7 murine macrophages. CCK8 test showed that rCpStefin below 50 μg/mL concentration did not affect cellular viability. Moreover, the NO production level determined by the Griess test was decreased. In addition, the secretion levels of IL-1β, IL-6, and TNF-α as measured by ELISA were decreased. Furthermore, it exerted anti-inflammatory activity by decreasing the production of proinflammatory cytokines and proinflammatory mediators, including IL-1β, IL-6, TNF-α, iNOS, and COX-2 at the gene transcription level, as measured by qRT-PCR. Therefore, Type I cystatin derived from *C. pisiformis* suppresses the LPS-Mediated inflammatory response of the intermediate host and is a potential candidate for the treatment of inflammatory diseases.

## 1. Introduction

Cysticercosis pisiformis is a parasitic disease caused by *C. pisiformis*, the metacestode of *T. pisiformis*, in rabbits and some other rodents, such as golden hamsters [1]. Cysticercosis pisiformis is distributed worldwide and has been reported in many countries in Europe, Asia, Africa, and South America [2,3] The intermediate host of *T. pisiformis* is rabbits and some other rodents, and the main definitive host is carnivores, such as dogs, cats, foxes and so on. With the increase in dog breeding, cysticercosis pisiformis has become one of the most common rabbit parasitic diseases, seriously affecting the rabbit breeding industry. *C. pisiformis* commonly parasitizes in the hepatic peritoneum, greater tomum, mesentery, and perirectum of intermediate hosts, while *T. pisiformis* parasitizes in the small intestine of definitive hosts [4]. Infection of *T. pisiformis* in a definitive host is prone to occur when the definitive host ingests internal organs infected with *C. pisiformis*. In the small intestine of definitive hosts, the parasite develops from its metacestode into an adult, which maturates and begins to discharge eggs after about 6–8 weeks. The egg-bearing segments are released into the environment along with the host’s feces [5]. The release of secreted excretory proteins is accompanied by the production of large amounts of cysteine protease inhibitors.

Inflammation heals and repairs tissues damage by signaling to the immune system, which holds a key role in the process of infection and injury [6,7]. However, the excessive inflammatory response may increase the risk of diabetes, cardiovascular disease, arthritis, allergies, psoriasis, and rheumatoid arthritis. Macrophages are important members of the innate immune system, which are capable of detecting anti-inflammatory and immunomodulators [8]. About 130 years ago, macrophages first were identified to be responsible for the innate immune system through the secretion of various inflammatory cytokines and chemokines, as well as phagocytosis [9]. Lipopolysaccharide (LPS) is an endotoxin present on the cell wall of gram-negative bacteria. In particular, LPS stimulates macrophages by means of toxin-like receptors (TLRs) and a series of downstream signaling cascades to trigger an increase in inflammatory mediators, such as nitric oxide (NO), prostaglandin E2 (PGE2), inducible nitric oxide synthase (iNOS), and cyclooxygenase-2 (COX-2), as well as the rise of proinflammatory cytokines, for example, tumor necrosis factor-alpha (TNF-α), interleukin-1β (IL-1β) and interleukin-6 (IL-6) [10,11,12,13]. Therefore, regulation of the expression of inflammatory mediators is of great significance for the treatment of some chronic diseases.

Numerous experimental and clinical research, as well as epidemiological data, have confirmed that certain parasites and their derived biological macromolecules can prevent or treat allergic and autoimmune diseases [14,15]. Parasites and their proteins can regulate the pathological damage caused by excessive immune response by stimulating the activation of immune cells and promoting or inhibiting the secretion of cytokines, so as to significantly alleviate the development of allergic and autoimmune diseases. Parasite-related proteins are an important class of immunomodulators, and a great deal of research in recent years has shown that secretory excretions of parasites can modulate the phenotype and function of immune cells [16,17,18,19,20]. It can interfere with all aspects of host immunity, and cysteine protease inhibitors are the main immunomodulators. Secretory excretory antigens derived from parasites may be considered as drugs for the treatment of inflammation-related diseases, such as autoimmune, metabolic disorders, and chronic diseases. Epidemiological studies have indicated that helminth infection is negatively correlated with inflammatory bowel disease (IBD) or allergies [21,22,23]. Multiple experimental studies in mice have summarized this negative correlation and have shown that concurrent helminth infections can ameliorate the diseases [24]. Therefore, further research on secreted excretory antigens of parasite origin has important implications for the future treatment and control of inflammatory diseases [25].

Proteases are classified into four types based on their hydrolysis mechanism and active structure: aspartic proteases, matrix metalloproteinases (MMPs), serine proteases, and cysteine proteases (CPs). Among them, CPs are intracellular proteases in lysosomes, which are involved in extracellular matrix (ECM) degradation, apoptosis, immunomodulation, and angiogenesis. The functions and activities of cysteine proteases are tightly regulated by endogenous inhibitors, namely the cysteine protease inhibitor (CPI) superfamily. CPI superfamily is widely present in organisms including plants, protozoa, mammals, bacteria, viruses, and parasites, and is able to tightly and reversibly bind to cysteine proteases [26]. Thus, the CPI superfamily is also involved in a variety of physiological and pathological processes by inhibiting the activity of CPs [27]. Since the discovery, the CPI family has evolved and grown into a superfamily of genes, which are divided into four main families including Family 1 (Stefins Family), Family 2 (Cystatins Family), Family 3 (Kininogens Family), and Family 4 (Fetuin Family) [20]. CPI derived from parasites can not only inhibit the activity of CPs from the host, but also affect antigen presentation, regulate cytokine secretion, and up-regulate NO synthesis, thereby creating conditions for the survival of parasites in the host [18,19,20].

In recent years, more and more scientific research has confirmed that CPI derived from different species of parasites were immunoregulatory molecules of intermediate host and were effective against different types of inflammatory bowel disease as well as allergic asthma [28,29,30]. However, the role of Stefin derived from *C. pisiformis* (Cpstefin) in host immune response has not been clarified in detail. Therefore, the purpose of the present study was to evaluate the anti-inflammatory effect of recombinant Stefin derived from *C. pisiformis* on LPS-induced macrophages. An anti-inflammatory model of rCpstefin was proposed after an investigation of the anti-inflammatory effect of recombinant Cpstefin (Figure 1).

## 2. Materials and Methods

### 2.1. Production of Recombinant Cpstefin and Removal of Endotoxin

The *Escherichia coli* strain BL21(DE3) harboring recombinant plasmid pGEX-4T-CpStefin preserved in our laboratory was activated and was expanded to an optical density (OD) value of 0.6–1.0 followed by induction using IPTG with a final concentration of 1 mmol. After being shaken at 16 °C at 180 r/min overnight, the cultured product was collected and centrifuged at 10,000 rpm for 15 min followed by ultrasonic disruption on ice (work for 3 s, interval for 4 s, repeat for 20 min). Then, the expression of rCpstefin in the supernatant was determined by SDS-PAGE electrophoresis. Afterwards, rCpstefin was purified by GST•Bind^TM^ resin, identified by SDS-PAGE electrophoresis, and the protein concentration was determined by using a Bradford Protein Concentration Assay Kit (Beyotime Biotechnology, Shanghai, China).

The principle of liquid-phase separation was applied to remove endotoxin from rCpstefin using Triton X-114 (Sigma Aldrich, Darmstadt, Germany) according to instruction of manufacture, followed by filtration with a 0.22 μm filter. Afterward, the level of endotoxin in rCpstefin was measured using a ToxinSensor™ Chromogenic LAL Endotoxin Assay Kit (GenScript, Piscataway, NJ, USA). The level of endotoxin rCpstefin less than 1 EU/mg (<0.1 ng/mg) was acceptable during cell culture and was applied to treat RAW264.7 cells in the following experiment [31].

### 2.2. Cell Culture and Treatments

RAW264.7 cell was provided by the Veterinary Public Health Laboratory of Gansu Agricultural University. Cell culture solution containing RPMI 1640 cell culture medium (Gibco, ThermoFisher Biochemical Product Co., Ltd., Beijing, China). A total of 10% fetal bovine serum (BI, Shanghai XP Biomed Co., Ltd., Shanghai, China) and 1% penicillin-streptomycin liquid (Solarbio, Beijing, China) was prepared and RAW264.7 cells were resuscitated.

Resuscitated RAW264.7 cells then were transferred into the T25 cell culture bottle (Servicebio, Wuhan, China) and 4 mL cell culture solution was added into the cell culture bottle followed by a gentle mixture. RAW264.7 cells were cultured in a cell culture incubator (ESCO, Shanghai, China) injected with 5% CO_2_ at 37 °C, and were passaged at least three times. Cultured RAW264.7 cells with a cell density above 80% can only be used for subsequent experiments. Then, the cells were seeded into 96-well plates or 12-well plates according to the specific requirements of further experiments.

### 2.3. Cell Viability Assay

The effect of rCpStefin and LPS on the cellular activity of mouse macrophage RAW264.7 was detected by a cell proliferation and cytotoxicity assay kit (Cell Counting Kit-8, Beyotime Biotechnology, China). Well growing RAW264.7 cells were inoculated into 96-well plates at a cell density of 1 × 10^4^ per well and then were cultured for 24 h. Afterward, the original culture medium was replaced by a complete culture medium added with rCpStefin reaching from 5 to 50 μg/mL and cells in the original culture medium in the experimental group were substituted with complete culture medium mixed with rCpStefin reaching from 5 to 50 μg/mL in the presence of LPS (500 ng/mL), respectively. Cell cultures with the absence of rCpStefin were set as blank control in both groups and cell cultures only in the presence of LPS were deemed as the positive control. After incubation for 24 h, the culture medium in both the control and experimental group was removed and then 10% CCK8 reagent (CCK8 and cell culture medium) was added into cell cultures in each well followed by incubation at 37 °C for 45 min in the dark. Finally, the absorbance at the wavelength of 450 nm in each well was detected by a SpectraMax iD3 Multi-functional microplate reader (Meigu Molecular Instrument Co., Ltd., Shanghai, China).

### 2.4. Evaluation of NO by Griess Test

The classical Griess reaction was used to determine the level of NO secretion in RAW264.7 cell culture supernatants after treatment with rCpStefin under LPS induction utilizing a Nitrite Assay Kit purchased from Beyotime Biotechnology. Well growing RAW264.7 cells were seeded in 12-well plates at a cell density of 2 × 10^6^ per well, and the original culture medium was discarded when the cell density reached 80%. DMEM culture medium together with rCpStefin (5 μg/mL, 10 μg/mL, 20 μg/mL) in the presence of LPS (500 ng/mL) was added into wells of plates. The RAW264.7 cells in the negative control group treated with TpStefin (20 μg/mL) and the same cells in the positive control group incubated with LPS (500 ng/mL) were included. After incubation for 24 h, the total NO content of the cell supernatant was detected according to the manufacturer’s manual. Brifely, the standard substance NaNO_2_ was firstly diluted into 2 μmol/L, 5 μmol/L, 10 μmol/L, 20 μmol/L, 40 μmol/L, 60 μmol/L, and 80 μmol/L using DMEM culture solution, respectively. In succession, the cell culture supernatant post centrifugation (12,000 rpm for 5 min) and standard substance were incubated at 37 °C for 30 min after being added with nicotinamide adenine dinucleotide phosphate (NADPH), flavin adenine dinucleotide (FAD) and nitrate reductase in turn. Then, lactate dehydrogenase (LDH) mixed with LDH buffer was added and incubated at 37 °C for 30 min. Furthermore, Griess Reagent I and Griess Reagent II were added and incubated for 10 min at room temperature. At last, absorbance at 540 nm was determined using SpectraMax iD3 Multi-functional microplate reader (Meigu Molecular Instrument Co., Ltd., Shanghai, China) and the nitrite concentration of each sample was calculated utilizing a calibration curve based on the absorbance values of a standard substance.

### 2.5. ELISA Test Cytokine Secretion Levels

The effects of rCpStefin on the secretion levels of cytoproinflammatory factors IL-1β, IL-6, and TNF-α in mouse macrophage RAW264.7 induced by LPS were tested using an ELISA kit purchased from Shanghai Enzyme-linked Biological Company. Well growing RAW264.7 cells were seeded at a cell density of 2 × 10^6^ per well in 12-well plates, and the original culture medium was discarded when the cell density reached 80%. Cell culture medium with different concentrations (5 μg/mL, 10 μg/mL, 20 μg/mL) of rCpStefin in the presence of LPS (500 ng/mL) were added and the RAW264.7 cells in negative control group treated with TpStefin (20 μg/mL) and same cells in positive control group incubated with LPS (500 ng/mL) were also introduced. After incubation for 24 h, the cell supernatant was collected and the cytokine secretion levels including IL-1β, IL-6, and TNF-α in cell supernatant were quantified by using a sandwich ELISA kit of Shanghai Enzyme-linked Biological Company. Briefly, before the test, the kit and samples were brought to room temperature for use. At first, 50 μL standard product with different concentrations and 10 μL sample diluted with 40 μL sample diluent were added into corresponding wells. Then, 100 μL enzyme-labeled reagent was mixed into each well followed by incubation for 60 min at 37 °C. After rinsing with 1×washing solution five times, 50 μL of color developer A and B were added to each well successively and gently shaken to develop color at 37 °C for 15 min. After color development, 50 μL termination solution was added to each well to terminate the reaction. The absorbance of each hole at 450 nm wavelength was measured by the SpectraMax iD3 Multi-functional microplate reader (Meigu Molecular Instrument Co., Ltd., Shanghai, China).

### 2.6. Detection of Transcript Levels of Cytokines by Reverse Transcription Real-Time PCR

The mRNA level of IL-1β, IL-6, TNF-α, iNOS, COX-2, and the control gene GAPDH was detected using the reverse transcription kits HiScript III RT SuperMix for qPCR (Vazyme Biotech Co., Ltd., Nanjing, China) and ChamQ Universal SYBR qPCR Master Mix (Vazyme Biotech Co., Ltd., Nanjing, China). Well growing RAW264.7 cells were seeded in 12-well plates at a cell density of 2 × 10^6^ per well, and the original culture medium was discarded when the cell density reached 80%. Cell culture medium with different concentrations (5 μg/mL, 10 μg/mL, 20 μg/mL) of rCpStefin in the presence of LPS (500 ng/mL) were added and the RAW264.7 cells in negative control group treated with TpStefin (20 μg/mL) and same cells in positive control group incubated with LPS (500 ng/mL) were also introduced. After incubation for 24 h, cell precipitation from samples was collected and RNA from cell precipitation was extracted using Trizol reagent (Lanbolide Biotechnology Co. Ltd., Chengdu, China). The extracted RNA was reverse-transcribed using HiScript III RT SuperMix for qPCR (Vazyme Biotech Co., Ltd., Nanjing, China), and the RNA concentration was measured by an ultraviolet spectrophotometer. The mRNA levels of cellular pro-inflammatory factors IL-1β, IL-6, TNF-α, iNOS, and COX-2 were determined by the SYBR Green fluorescent dye method using GAPDH as an endogenous reference gene, and the experimental data were summarized and analyzed by the 2^−ΔΔCT^ relative quantification method. Primers for gene amplification are shown in Table 1.

### 2.7. Statistical Analysis 

All experiments including cell viability, NO content, levels of secretion of inflammatory factors and mediators, and transcript levels of inflammatory factors and mediators were conducted independently in three replicates. The mean value and standard deviation (SD) of three replicates in each experiment were calculated and then the mean ± SD of three replicates were applied to represent data in each experiment. The statistical significance between quantitative values including cell viability, NO content, levels of secretion of inflammatory factors and mediators, and transcript levels of inflammatory factors and mediators were calculated by one-way analysis of variance (ANOVA), followed by Dunnett’s multiple comparison test comparing all the concentrations of rCpStefin (5 μg/mL, 10 μg/mL, 20 μg/mL) with the control utilizing GraphPad Prism 8 (GraphPad Software, version 8, San Diego, CA, USA). Differences in mean values of each experiment were deemed as statistically significant when the *p*-value was less than 0.05. 

## 3. Results and Analysis

### 3.1. Expression and Purification of rCpStefin Proteins

The recombinant plasmid pGEX-4T-CpStefin was transformed into *E. coli* strain BL21(DE3) cells and the expression of rCpStefin induced by IPTG were analyzed by SDS-PAGE electrophoresis. The size of rCpStefin was about 36 kDa and mainly distributed in the supernatant (Appendix A), indicating that the size of rCpStefin protein corresponded to the target protein and rCpStefin was solubly expressed. Furthermore, the rCpStefin was purified by GST•Bind™ resin, and the single purpose band of 36 kDa was obtained by SDS-PAGE electrophoresis, which denoted that highly purified rCpStefin was obtained (Appendix A).

### 3.2. CpStefin Has No Effect on LPS-Induced Activity in RAW264.7 Cells

In order to optimize the rCpStefin concentration applied for the subsequent tests and evaluate the influence of rCpStefin, a CCK8 kit was utilized to determine the cell toxicity. rCpStefin reaching 5 to 50 μg/mL was supplemented to the culture media. The results of this experiment demonstrated that incubation of RAW264.7-derived macrophages with rCpStefin reaching from 5 to 20 μg/mL for 24 h did not change the cell survival rate (Figure 2A). In the experimental group, there was no significant difference regarding the cell viability values of RAW264.7-derived macrophages treated with rCpStefin, LPS, and the co-induction of rCpStefin (5~50 μg/mL) and LPS for 24 h, compared with the control group (Figure 2B). Thus, it is shown that rCpStefin at concentrations of 5 μg/mL, 10 μg/mL, and 20 μg/mL can be safely used in subsequent experiments.

### 3.3. rCpStefin Inhibits LPS-Induced NO Production in RAW264.7 Cells

NO is an important molecular marker of macrophage-mediated inflammatory response. Therefore, the effects of different concentrations of rCpStefin on LPS-induced NO production by RAW264.7 cells were examined. As shown in Figure 3, LPS significantly increased the production of NO in RAW264.7 cells (*p* < 0.0001). Compared with the LPS positive control group, rCpStefin inhibited NO production in RAW264.7 cells in a dose-dependent manner.

### 3.4. rCpStefin Inhibits LPS-Induced Cytokine Production by RAW264.7

IL-1β, IL-6, and TNF-α are important pro-inflammatory cytokines. In order to detect the effects of rCpStefin on the production and release of IL-1β, IL-6, and TNF-α from RAW264.7 induced by LPS, levels of IL-1β, IL-6, and TNF-α in the culture supernatant of each group were detected by double-antibody sandwich ELISA kit. Compared with the blank control group, the levels of anti-inflammatory cytokines IL-1β (Figure 4A), IL-6 (Figure 4B), and TNF-α (Figure 4C) secreted by RAW264.7 cells after stimulation by LPS alone increased significantly after 24 h (*p* < 0.0001). Compared with the LPS-positive control group, the levels of pro-inflammatory cytokines IL-1β, IL-6, and TNF-α secreted by RAW264.7 cells were significantly inhibited by rCpStefin and decreased in a dose-dependent manner (Figure 4A–C). However, the treated group with 5 μg/mL rCpStefin showed a significant decrease in IL-1β (*p* < 0.05) and IL-6 (*p* < 0.05) secretion except in TNF secretion (Figure 4A–C).

### 3.5. rCpStefin Inhibits mRNA Levels of Cytokine Production by RAW264.7 under LPS Induction

To confirm the LPS-Mediated release of pro-inflammatory cytokines and inflammation-associated molecules, the mRNA expression of the above mentioned molecules was assessed by qRTPCR. As illustrated in Figure 5 and Figure 6, compared with the blank control group, mRNA levels of cytokines including IL-1β (*p* < 0.0001), IL-6 (*p* < 0.0001) and TNF-α (*p* < 0.0001) were remarkable increase in LPS alone-stimulated RAW264.7 cells after 24 h, as well as the mRNA levels of inflammation-related molecules COX-2 (*p* < 0.0001) and iNOS (*p* < 0.001). Compared with the LPS-positive control group, both the mRNA levels of cytokines (IL-1β, IL-6, and TNF-α) and the mRNA levels of inflammation-related molecules (iNOS and COX-2) were significantly suppressed and reduced in a dose-dependent manner in the rCpStefin treated group (Figure 5 and Figure 6).

## 4. Discussion

In recent years, it has been shown that helminths and their proteins can suppress the host immune response, so those proteins have the potential to prevent or treat allergic and autoimmune diseases by modulating immune cells, such as macrophages [32]. During infection by helminths, the host immune response is modulated by altering the pattern of activation during macrophage “polarization” towards effector function [24]. Macrophages are classified into classically activated macrophages (M1-type macrophages) and alternatively activated macrophages (M2-type macrophages) based on their function and phenotype after polarization [33]. M1-type macrophages induced by lLPS and γ-interferon (IFN-γ) are characterized by the production of high levels of pro-inflammatory factors that promote Th1 and Th17 cell differentiation and activation and further exacerbate inflammation and tissue damage [34,35]. M2 macrophages are induced by the interleukins IL-4 and IL-13, which secrete large amounts of IL-10 and the transforming growth factor TGF-β and also promote high expression of mannose receptor (CD206), arginase (Arg-1), and CD163, which exhibit potent anti-inflammatory properties [36,37,38]. Therefore, modulation of macrophage polarization toward M2-type macrophages with anti-inflammatory properties has become one of the effective ways to prevent and treat allergic and autoimmune diseases. *T. spiralis* rTs-Cys was co-incubated with mouse bone marrow-derived macrophages (BMDMs) in vitro to observe whether it could regulate macrophage polarization toward M2-type macrophages with anti-inflammatory properties and inhibit M1-type macrophage activation, and it has been preliminarily demonstrated that rTs-Cys could regulate macrophage polarization [39]. Exosomal microRNA let-7 from *T. pisiformis* metacestodes can induce M2 macrophage polarization via targeting C/EBP δ [40].

Bacterial lipopolysaccharide-induced macrophages are a common pathogenic molecular pattern [25]. LPS causes inflammation by stimulating intercellular signaling pathways, including transcription factor nuclear factor-κB (NF-κB) and mitogen-activated protein kinase (MAPK) [41]. Both NF-κB and MAPK pathways were considered to be classical pathways regulating inflammation for the reason that they initiated the secretion of inflammation-associated molecules, regulated oxidative stress, and accelerated the progression of inflammation [41]. MAPK was an important transmitter of signals from the cell surface to the interior of the nucleus, where signals were transmitted through a phosphorylation cascade [42]. MAPK was divided into four subfamilies: extracellular regulated protein kinases (ERK), p38, c-Jun N-terminal kinase (JNK), and ERK5. These pathways are named after them. For example, the MAPK pathway using JNK is called the JNK pathway [43]. ERK is an important mediator in signal transduction in response to cellular stress and various tissue injuries [44]. p38 (MAPK) protein kinase affects various intracellular reactions, mediating inflammation, apoptosis, and so on [45]. The JNK family is a key molecule in cell signal transduction induced by various stressors, which can induce the production of inflammatory factors [46]. MAPK is phosphorylated by external stimuli, such as LPS, leading to NF-κB phosphorylation, which causes an inflammatory response. NF-κB is an important nuclear transcription factor in the cell [47]. It is involved in the body’s inflammatory response and can regulate apoptosis, stress response, NF-κB overactivation. Therefore, it is related to many human diseases, such as rheumatoid arthritis, heart and brain diseases, and inflammatory changes. Thus, drugs capable of inhibiting the NF-κB signal transduction pathway may be useful for the treatment of these diseases. Overactivation of NF-κB increases the expression of inducible nitric oxide synthase (iNOS) and cyclooxygenase-2 (COX-2) in the synthesis of inflammatory factors [48]. iNOS is an inducible nitric oxide synthase. It is a key enzyme that is expressed under inflammatory or other stimulating conditions and is capable of producing high levels of NO under these conditions [49]. NO has been recognized as one of the most versatile players in the immune system, it is involved in the pathogenesis of inflammatory responses, infectious diseases, tumors, and chronic degenerative diseases [50]. COX-2 is a key enzyme that catalyzes the conversion of arachidonic acid into prostaglandins, and inflammatory injury mainly stimulates immune cells and induces the production of COX-2 [51]. In vitro animal cell systems, these pathways can activate the molecular mechanism of anti-inflammatory effects. *C. pisiformis*-derived novel-miR1 exerts its regulatory effects by inhibiting the expression of TLR2, thereby modulating the production of pro-inflammatory factors via the NF-κB pathway [52]. Cystatin I from *Fasciola gigantica* inhibited the expression of NF-κB signaling pathway-related molecules in LPS-activated RAW264.7 cells [25]. Marine-derived fats and fatty acids have also been shown to reduce endotoxin-induced immune responses by inhibiting phosphorylation of NF-κB and MAPK signal transduction pathways [53]. Extracts from *Passiflora foetida* and MAPK inhibitors not only could reduce suppressed effects of phosphorylation of JNK, p38, and ERK1/2 on LPS-induced NO, IL-6, and TNF-α production but also could suppress IκBα phosphorylation on NO production [54]. Schisantherin A and coptisine from *Coptis chinensis* could suppress LPS-stimulation of MAPK phosphory-lation by specific inhibitors [55]. In the field of parasites, especially tapeworms, there were few studies about the regulation of host immune response. Combined with the above LPS-induced inflammation model, Cpstefin is speculated to have anti-inflammatory effects by inhibiting the phosphorylation of NF-κB and MAPK pathways. 

In this study, in order to exclude the cytotoxic effects of the rCpstefin, cell viability was determined by the CCK8 assay. The results showed that rTpStefin reaching from 5 to 20 μg/mL for 24 h did not alter the cell survival rate (Figure 2A). Co-incubation of LPS with rCpstefin at concentrations of 5, 10, 20, and 50 µg/mL did not alter cell viability in RAW 264.7 cells (Figure 2B). This indicates that rCpstefin can be safely used at the concentrations tested without cytotoxic effects.

To evaluate the anti-inflammatory effects of rCpstefin on the secretion of inflammatory factors and NO release resulting in RAW264.7 cells, double-antibody sandwich ELISA and Griess assay were used, respectively. As expected, rCpstefin reduced IL-1β, IL-6, TNF-α (Figure 4A–C), and NO (Figure 3) secretion in LPS-induced macrophage culture supernatants in a dose-dependent manner. The mRNA levels of pro-inflammatory cytokines (IL-1β, IL-6, and TNF-α) and inflammatory mediators (iNOS and COX-2) were analyzed by qRT-PCR (Figure 5A–C and Figure 6A,B). It was shown that rCpstefin had a potential modulatory effect on macrophage inflammation, which suggested that rCpstefin could be further applied to the study of autoimmune diseases characterized by a significant increase in IL-1β, IL-6, and TNF-α, such as inflammatory bowel inflammation. Although this study indicated that rCpstefin had the potential anti-inflammatory effects by regulating cytokine levels, whether regulation of cytokine secretion by rCpstefin is related to the polarization of macrophages and the signaling pathway of the polarization of macrophages remains unclear. Previous experiments have cloned and expressed the Stefin gene of *Cysticercus Pisiformis* cysteine protease inhibitor and prepared its polyclonal antibody, which provides conditions for elucidating the anti-inflammatory properties of the Stefin family of Cysticercus Pisiformis. The regulatory mechanisms of the anti-inflammatory effect of rCpstefin and the potential of rCpstefin as a therapeutic agent for the treatment of inflammatory diseases will be interesting and meaningful research in our future study. 

## 5. Conclusions

In conclusion, this assay demonstrated that rCpstefin reduced the production of pro-inflammatory cytokines and mediators in LPS-activated mouse RAW 264.7 cells and inhibited LPS-mediated inflammatory responses in RAW 264.7 cells, thereby exerting anti-inflammatory properties. These results confirmed that rCpstefin affected the production of pro-inflammatory cytokines (IL-6, TNF-α, and IL-1β) and mediators (NO, iNOS, and COX-2) at the level of gene transcription, and secretion. The results of the present study will be useful for further investigation of the immunoregulatory mechanisms and anti-inflammatory potential of cysteine protease inhibitors of *T. pisiformis.*

## Figures and Tables

**Figure 1 microorganisms-12-00850-f001:**
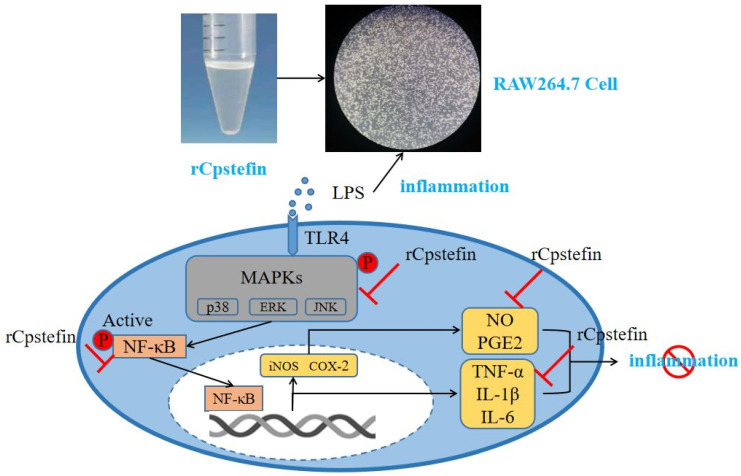
The anti-inflammatory activity model of rCpStefin in macrophages was proposed. The black lines represented the normal response and the red areas denoted the possible activity of rCpStefin.

**Figure 2 microorganisms-12-00850-f002:**
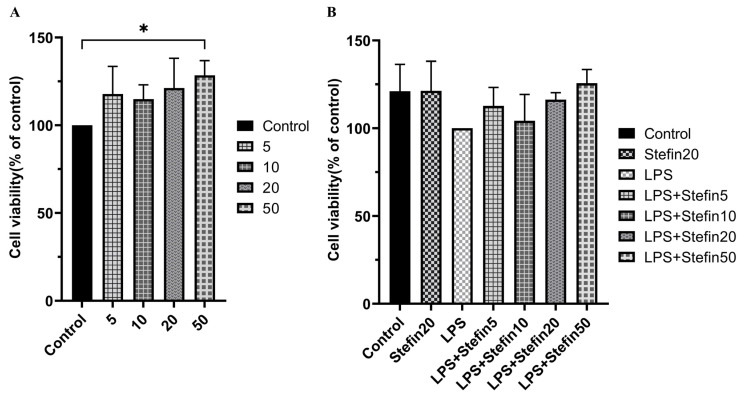
Effect of various concentrations of rCpStefin in the RAW264.7 cell viability. (**A**). Cell viabilities of RAW264.7 treated with various concentrations of rCpStefin under the condition of LPS (500 ng/mL) absence. *x*-axis denoted that RAW264.7 cell was treated with various concentrations of rCpStefin, and *y*-axis indicated the RAW264.7 cell viability. Asterisk and “ns” above the horizontal line mean the presence or absence of significant difference. One asterisk represents *p* < 0.05. (**B**). Cell viabilities of RAW264.7 treated with various concentrations of rCpStefin under the condition of LPS (500 ng/mL) presence. *x*-axis denoted that the RAW264.7 cell was treated with various concentrations of rCpStefin under the condition of LPS (500 ng/mL) presence, and *y*-axis indicated the RAW264.7 cell viability. Asterisk and “ns” above the horizontal line mean the presence or absence of significant difference. One asterisk represents *p* < 0.05.

**Figure 3 microorganisms-12-00850-f003:**
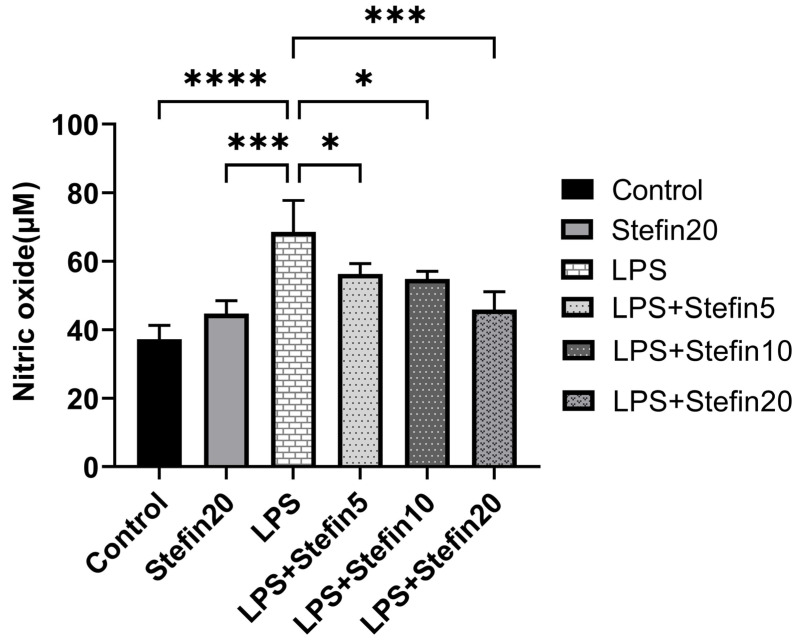
Effects of rCpStefin (5, 10, and 20 μg/mL) on NO production in the RAW264.7 cells activated with 500 ng/mL LPS. *x*-axis denoted that the RAW264.7 cell was treated with various concentrations of rCpStefin under the condition of LPS (500 ng/mL) presence and *y*-axis indicated NO production. Asterisk and “ns” above the horizontal line mean the presence or absence of significant difference. One asterisk, three asterisks, and four asterisks, respectively, represent *p* < 0.05, *p* < 0.001, and *p* < 0.0001.

**Figure 4 microorganisms-12-00850-f004:**
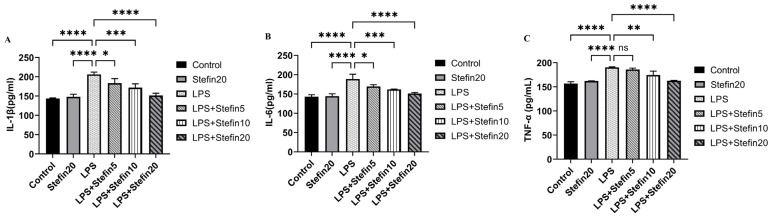
Effects of rCpStefin (5, 10, and 20 µg/mL) on secretion of proinflammatory cytokines in RAW264.7 cells activated with 500 ng/mL LPS. (**A**). Secretion of IL-1β in RAW264.7 cells activated with 500 ng/mL LPS. *x*-axis denoted that the RAW264.7 cell was treated with various concentrations of rCpStefin under the condition of LPS (500 ng/mL) presence and *y*-axis indicated that the secretion of IL-1β in cell supernatant was detected. Asterisk and “ns” above the horizontal line mean the presence or absence of significant difference. One asterisk, three asterisks, and four asterisks, respectively, represent *p* < 0.05, *p* < 0.001, and *p* < 0.0001. (**B**). Secretion of IL-6 in RAW264.7 cells activated with 500 ng/mL LPS. *x*-axis denoted that the RAW264.7 cell was treated with various concentrations of rCpStefin under the condition of LPS (500 ng/mL) presence and *y*-axis indicated that the secretion of IL-6 in cell supernatant was detected. Asterisk and “ns” above the horizontal line mean the presence or absence of significant difference. One asterisk, three asterisks, and four asterisks, respectively, represent *p* < 0.05, *p* < 0.001, and *p* < 0.0001. (**C**). Secretion of TNF-α in RAW264.7 cells activated with 500 ng/mL LPS. *x*-axis denoted that the RAW264.7 cell was treated with various concentrations of rCpStefin under the condition of LPS (500 ng/mL) presence and *y*-axis indicated that the secretion of TNF-α in cell supernatant was detected. Asterisk and “ns” above the horizontal line mean the presence or absence of significant difference. Two asterisks, four asterisks, and ns, respectively, represent *p* < 0.01, *p* < 0.0001, and nonsignificant.

**Figure 5 microorganisms-12-00850-f005:**
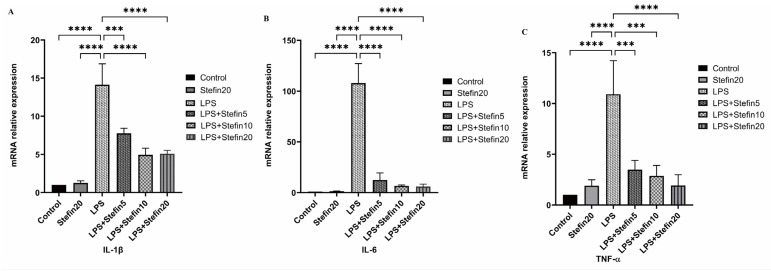
Effects of rCpStefin (5, 10, and 20 µg/mL) on the mRNA level of proinflammatory cytokines in the RAW264.7 cells activated with 500 ng/mL LPS. (**A**). The mRNA level of IL-1β in the RAW264.7 cells was activated with 500 ng/mL LPS. *x*-axis denoted that the RAW264.7 cell was treated with various concentrations of rCpStefin under the condition of LPS (500 ng/mL) presence and *y*-axis indicated that transcript levels of IL-1β in cell sedimentation. Asterisk and “ns” above the horizontal line mean the presence or absence of significant difference. Three asterisks, and four asterisks, respectively, represent *p* < 0.001, and *p* < 0.0001. (**B**). The mRNA level of IL-6 in the RAW264.7 cells was activated with 500 ng/mL LPS. *x*-axis denoted that the RAW264.7 cell was treated with various concentrations of rCpStefin under the condition of LPS (500 ng/mL) presence and *y*-axis indicated that transcript levels of IL-6 in cell sedimentation. Asterisk and “ns” above the horizontal line mean the presence or absence of significant difference. Four asterisks represent *p* < 0.0001. (**C**). The mRNA level of TNF-α in the RAW264.7 cells was activated with 500 ng/mL LPS. *x*-axis denoted that the RAW264.7 cell was treated with various concentrations of rCpStefin under the condition of LPS (500 ng/mL) presence and *y*-axis indicated transcript levels of TNF-α in cell sedimentation. Asterisk and “ns” above the horizontal line mean the presence or absence of significant difference. Three asterisks, and four asterisks, respectively, represent *p* < 0.001, and *p* < 0.0001.

**Figure 6 microorganisms-12-00850-f006:**
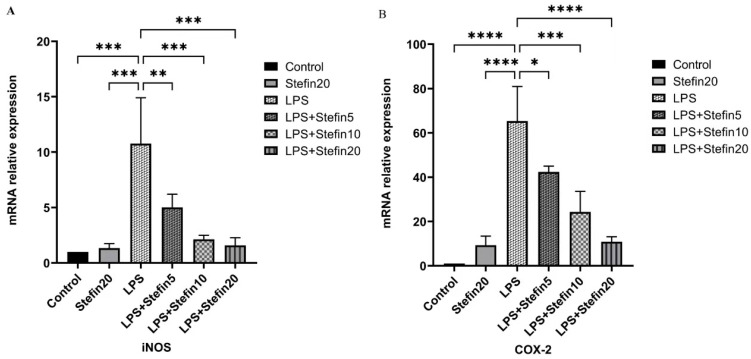
Effects of rCpStefin (5, 10, and 20 µg/mL) on the mRNA level of proinflammatory cytokines in the RAW264.7 cells activated with 500 ng/mL LPS. (**A**). The mRNA level of iNOS in the RAW264.7 cells was activated with 500 ng/mL LPS. *x*-axis denoted that the RAW264.7 cell was treated with various concentrations of rCpStefin under the condition of LPS (500 ng/mL) presence and *y*-axis indicated that transcript levels of iNOS in cell sedimentation. Asterisk and “ns” above the horizontal line mean the presence or absence of significant difference. Two asterisks, and three asterisks, respectively, represent *p* < 0.01, and *p* < 0.001. (**B**). The mRNA level of COX-2 in the RAW264.7 cells was activated with 500 ng/mL LPS. *x*-axis denoted that the RAW264.7 cell is treated with various concentrations of rCpStefin under the condition of LPS (500 ng/mL) presence and *y*-axis indicated that transcript levels of COX-2 in cell sedimentation. Asterisk and “ns” above the horizontal line mean the presence or absence of significant difference. One asterisk, three asterisks, and four asterisks, respectively, represent *p* < 0.05, *p* < 0.001, and *p* < 0.0001.

**Table 1 microorganisms-12-00850-t001:** Primer sequences used in RT-PCR analysis.

Gene	Primer Sequence from 5′ to 3′	Product Size (bp)
IL-1β	FW: TGC CAC CCT TTT GAC AGT GAT G	138
	RV: TGA TGT GCT GCT GCG AGA TT	
IL-6	FW: CCC CAA TTT CCA ATG CTC TCC	141
	RV: CGC ACT AGG TTT GCC GAG TA	
TNF-α	FW: CCC TCA CAC TCA GAT CAT CTT CT	61
	RV: GCT ACG ACG TGG GCT ACA G	
COX-2	FW: TGT GAC TGT ACC CGG ACT GG	233
	RV: TGC ACA TTG TAA GTA GGT GGA C	
iNOS	FW: CCC TTC CGA AGT TTC TGG CAG CAG	497
	RV: GGC TGT CAG AGC CTC GTG GCT TTG G	
GAPDH	FW: CGA CTT CAA CAG CGA CAC TCA C	119
	RV: CCC TGT TGC TGT AGC CAA ATT C	

## Data Availability

The data related to this study can be obtained by contacting the first and corresponding authors by email.

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
