# Peer review of "Type I Cystatin Derived from Cysticercus pisiformis—Stefins, Suppresses LPS-Mediated Inflammatory Response in RAW264.7 Cells"

_microorganisms, 2024, doi:10.3390/microorganisms12050850_

Round 1

Reviewer 1 Report

Comments and Suggestions for Authors

Dear authors,

Suggestions and comments follow:

- Line  1: Replace "Pisiformis" with "pisiformis". Remove a "-".

- Line 9: Replace "Cysticercosis" with "Cysticercus".

- Line 9-10: Replace "tapeworm disease caused by the cysticercosis of taenia pisiformis" with "tapeworm larvae of Taenia pisiformis".

- Line 13; 20-21; 23: Replace "Pisiformis" with "pisiformis".

- References are not in the appropriate format in the text. put it between [ ].

- All citations require review, including the numbering order.

- Line 93-106: Paragraph not necessary.

- Genus Cysticercus needs to be abbreviated in the text after the first mention.

- Line 238: supernant?

- Line 373: Replace "Trichinella" with "T.".

- Line 401; 404: Error! Reference source not found?

Reviewer 2 Report

Comments and Suggestions for Authors

The study is interesting and paves the way for new research. I have no comments except that it should be added to the bibliography : 1. C pisiformis- derived novel-miR1targets......., Chen G. et all., Front Immunol., 2023. 2. Characterisations of exosome-like vesicles derived from T. pisiformis cysticercus......., Wang et all., Parasit Vectors, 2020.

Reviewer 3 Report

Comments and Suggestions for Authors

The work brings an important topic to understanding Cysticercus pisiforms.

Some suggestions:

In the Title the name of the parasite was wrong Cysticercus Pisiformis. The correct one is Cysticercus pisiformis. With a small p.

The reference number must be in square brackets when there is more than one reference cited. As it is, it is confusing and does not comply with the magazine's standards. Check the instruction for authors.

In the methodology, in item 2.7 where the statistical analysis is described, it would be appropriate to state which analysis was used for each statistical test applied.

In the discussion of the work, a figure was placed on page 11 (figure 6). I believe that the explanation and this topic are more relevant in the introduction than in the discussion of the work. In general, we do not include figures in the discussion or explanatory schemes. Place the figure in the introduction with the appropriate explanation.
